# Full-Length Transcriptome Sequencing of the Scleractinian Coral *Montipora foliosa* Reveals the Gene Expression Profile of Coral–Zooxanthellae Holobiont

**DOI:** 10.3390/biology10121274

**Published:** 2021-12-05

**Authors:** Yunqing Liu, Xin Liao, Tingyu Han, Ao Su, Zhuojun Guo, Na Lu, Chunpeng He, Zuhong Lu

**Affiliations:** 1State Key Laboratory of Bioelectronics, School of Biological Science & Medical Engineering, Southeast University, Nanjing 210096, China; 230169123@seu.edu.cn (Y.L.); 220171811@seu.edu.cn (T.H.); 220212217@seu.edu.cn (A.S.); 220191941@seu.edu.cn (Z.G.); nlu@seu.edu.cn (N.L.); 2Guangxi Key Lab of Mangrove Conservation and Utilization, Guangxi Mangrove Research Center, Beihai 536000, China; liaox@mangrove.org.cn

**Keywords:** Scleractinia, zooxanthellae, holobiont, symbiosis, PacBio, transcriptomics

## Abstract

**Simple Summary:**

Coral bleaching (the disintegration of coral–zooxanthellae symbionts) is one of the important factors leading to coral death. However, we still lack an understanding of the mechanism of coral–zooxanthellae symbiosis. One of the reasons is the lack of reliable transcriptome sequence data. In this paper, through PacBio Sequel II sequencing technology polished with the Illumina RNA-seq platform, the *Montipora foliosa* transcriptome was obtained. The function and isoform of symbiosis-related genes were analyzed. This study provides a valuable resource for the study of coral symbiosis.

**Abstract:**

Coral–zooxanthellae holobionts are one of the most productive ecosystems in the ocean. With global warming and ocean acidification, coral ecosystems are facing unprecedented challenges. To save the coral ecosystems, we need to understand the symbiosis of coral–zooxanthellae. Although some Scleractinia (stony corals) transcriptomes have been sequenced, the reliable full-length transcriptome is still lacking due to the short-read length of second-generation sequencing and the uncertainty of the assembly results. Herein, PacBio Sequel II sequencing technology polished with the Illumina RNA-seq platform was used to obtain relatively complete scleractinian coral *M. foliosa* transcriptome data and to quantify *M. foliosa* gene expression. A total of 38,365 consensus sequences and 20,751 unique genes were identified. Seven databases were used for the gene function annotation, and 19,972 genes were annotated in at least one database. We found 131 zooxanthellae transcripts and 18,829 *M. foliosa* transcripts. A total of 6328 lncRNAs, 847 *M. foliosa* transcription factors (TFs), and 2 zooxanthellae TF were identified. In zooxanthellae we found pathways related to symbiosis, such as photosynthesis and nitrogen metabolism. Pathways related to symbiosis in *M. foliosa* include oxidative phosphorylation and nitrogen metabolism, etc. We summarized the isoforms and expression level of the symbiont recognition genes. Among the membrane proteins, we found three pathways of glycan biosynthesis, which may be involved in the organic matter storage and monosaccharide stabilization in *M. foliosa*. Our results provide better material for studying coral symbiosis.

## 1. Introduction

Corals are marine invertebrates within the class Anthozoa of the phylum Cnidaria that first appeared in the Cambrian approximately 535 million years ago. Many corals in the order Scleractinia are hermatypic, meaning that they are involved in building reefs [1]. They secrete calcium carbonate to form hard skeletons that become the framework of a reef [2]. Scleractinia is the key component of coral reef ecosystems, and it can extract calcium and carbonate to construct aragonite (biomineralization), which provides a habitat for other creatures. The biomineralization of Scleractinia cannot be achieved without the assistance of symbiotic algae [3,4]. Although some corals can catch plankton and small fish using stinging cells on their tentacles, scleractinian corals obtain most of their energy and nutrients from photosynthetic unicellular dinoflagellates of the genus *Symbiodinium* that live within their cells. The genus *Symbiodinium* is commonly known as zooxanthellae and provides coral its color. The main benefit of zooxanthellae is their ability to photosynthesize, thereby supplying corals with the products of photosynthesis, including glucose, glycerol, and amino acids, which corals can use for energy. Zooxanthellae benefit from a safe place to live and consume the polyps’ carbon dioxide, phosphate, and nitrogenous waste. Zooxanthellae also benefit corals by aiding in calcification, which is necessary for coral skeletons, and waste removal [5].

When the symbiont is exposed to environmental pressures, such as high temperatures and acidification, symbiotic algae are expelled or digested; this is otherwise known as coral bleaching [6,7,8]. With global warming and ocean acidification, bleaching has become the most serious threat for coral reefs. If the bleaching process continues, most of the coral ecosystems on Earth will completely disappear, and marine life will also face a serious survival crisis. However, the biological mechanisms of the bleaching process are still poorly understood. There is now a race between the study of symbiotic mechanisms and the process of bleaching. An important aspect of coral–zooxanthellae symbiosis is how they recognize one another, how they maintain their symbiosis [9], and how they exchange nutrients [10,11]. To investigate these questions, we need to conduct a comprehensive study of transcriptome expression in the symbiotic state of corals and zooxanthellae. Some coral and zooxanthellae genomes have been sequenced, providing a valuable resource for studying coral–dinoflagellate symbiosis mechanisms [12,13,14,15,16,17]. However, the currently published transcriptome is based on Illumina’s next-generation sequencing. Due to the short-read length, the quality of the full-length transcript, assembled based on the assembly algorithm, is relatively low. The advantage of full-length transcript sequencing methods is that they can offer longer read lengths than the second-generation sequencing (SGS or NGS) technologies, thereby complementing the NGS technology.

In our study, PacBio and Illumina sequencing were combined for the transcriptome study of symbiosis. The main goal of this study was to annotate the function of the genes in *M. foliosa*, to study alternative splicing, and to reveal the genes involved in symbiont identification and nutrient exchange.

## 2. Materials and Methods

### 2.1. Specimen Collection and Coral Culture System

*M. foliosa* in this study were collected from the Xisha Islands in the South China Sea (latitude 15°40′–17°10′ N, longitude 111°–113° E).

The coral samples were cultured in our laboratory coral tank with conditions conforming to their habitat environment. All the *M. foliosa* samples were raised in a RedSea^®^ tank (redsea575, Red Sea Aquatics Ltd., London, UK) at 26 °C and 1.025 salinity (Red Sea Aquatics Ltd., London, UK). The physical conditions of the coral culture system were as follows: Three coral lamps (AI^®^, Red Sea Aquatics Ltd., London, UK), a protein skimmer (regal250s, Reef Octopus), a water chiller (tk1000, TECO Ltd., Port Louis, Mauritius), two wave devices (VorTechTM MP40, EcoTech Marine Ltd. Bethlehem, PA, USA), and a calcium reactor (Calreact 200, Reef Octopus).

### 2.2. Sampling and RNA Extraction

All the RNA extraction procedures followed the manufacturer’s instructions. The total RNA was isolated with TRIzol LS Reagent (Thermo Fisher Scientific, 10296028, Waltham, MA, USA) and treated with DNase I (Thermo Fisher Scientific, 18068015). The high-quality mRNA was isolated with a FastTrack MAG Maxi mRNA Isolation Kit (Thermo Fisher Scientific, K1580-02). The samples were separated from healthy *M. foliosa* to ensure that enough high-quality RNA (>10 µg) could be obtained for a full-length cDNA transcriptome library. To support the accuracy and credibility of the data, we used three sample repetitions for the library construction and sequencing. Each sample was collected from an independent colony.

### 2.3. Library Preparation and Sequencing

#### 2.3.1. PacBio Library Preparation and Sequencing

Before the library construction, the quality of the total RNA had to be tested. Agarose gel electrophoresis was used to analyze the degree of degradation of the RNA and whether it was contaminated. A NanoDrop Nucleic Acid Quantifier was used to detect the purity of the RNA (OD260/280 ratio), a Qubit RNA assay was used to quantify the RNA concentration accurately, and an Agilent 2200 TapeStation was used to accurately detect the integrity of the RNA. A Clontech SMARTer^®^ PCR cDNA Synthesis Kit (Clontech Laboratories, Mountain View, CA, USA; 634926) and the BluePippin Size Selection System protocol, as described by Pacific Biosciences (PN 100-092-800-03), were used to prepare the Iso-Seq library according to the Isoform Sequencing protocol (Iso-Seq).

#### 2.3.2. Illumina Library Preparation and Sequencing

The illumine library was prepared according to Illumina (NEB, Ipswich, MA, USA) official NEBNext UltraTM RNA Library Prep Kit (E7530L) protocol. After the sample was qualified, the eukaryotic mRNA was enriched with magnetic beads with Oligo (dT). Subsequently, a fragmentation buffer was added to break the mRNA into short fragments. Using the mRNA as a template, one-stranded cDNA was synthesized with six-base random primers (random hexamers), and then, buffer, dNTPs, DNA polymerase I, and RNase H were added to synthesize the two-stranded cDNA. The double-stranded cDNA was purified with AMPure XP beads and added with poly-A tails and sequencing adapters then selected by fragment size with AMPure XP beads. The above cDNA was amplified by PCR and purified by AMPure XP beads to obtained final library, which was used for sequencing on the Illumina HiSeq X Ten platform for generating 150 bp paired-end reads.

### 2.4. Bioinformatics

#### 2.4.1. Data Processing

Sequence data were processed using SMRTlink 7.0 software [18]. A circular consensus sequence (CCS) was generated from the subread BAM files, parameters: min_length:50, max_drop_fraction:0.8, no_polish:TRUE, min_zscore:-9999.0, min_passes:1, min_predicted_accuracy:0.8, and max_length: 15000. The CCS.BAM files were output and were then classified into full-length and non-full-length reads using lima and removing polyA using refine. The full-length fasta files produced were then fed into the cluster step, which performed isoform-level clustering (ICE), followed by final Arrow polishing, hq_quiver_min_accuracy 0.99, bin_by_primer false, bin_size_kb 1, qv_trim_5p 100, and qv_trim_3p 30.

#### 2.4.2. Error Correction Using Illumina Reads

Additional nucleotide errors in the consensus reads were corrected using the Illumina RNA-seq data with LoRDEC software [19]. The full-length transcriptome was calibrated multiple times to improve data accuracy: (1) The same template was sequenced multiple times to perform in-hole correction for zero-mode waveguide holes to obtain a high-quality CCS; (2) cluster multi-copy transcriptional sequencing data were used to remove redundancy, perform inter-hole correction of the zero-mode waveguide holes, and obtain cluster consensus sequences; (3) Arrow software was used to calibrate the cluster consensus sequence using a non-full-length sequence and to establish a polished consensus sequence; and (4) the polished consensus sequence of the second-generation data was refined by LoRDEC software [19] to further improve the sequencing accuracy. LoRDEC uses the data of the second generation with high accuracy to correct the data of the third generation by PacBio. It adopts the mode of mixed error correction, with high accuracy and fast speed. Compared to correction software such as LSC and PacBioToCA, the accuracy of the LoRDEC’s correction is significantly improved. Any redundancy in the corrected consensus reads was removed using CD-HIT [20] (-c 0.95 -T 6 -G 0-aL 0.00 -aS 0.99) to obtain final transcripts for subsequent analysis. The software uses the heuristic algorithm to quickly find the highly similar fragments among sequences.

#### 2.4.3. Gene Functional Annotation

Gene functions were annotated based on the following databases: NR (NCBI non-redundant protein sequences) [21]; NT (NCBI non-redundant nucleotide sequences); Pfam (protein family); KOG/COG [22] (Clusters of Orthologous Groups of proteins); Swiss-Prot [23] (a manually annotated and reviewed protein sequence database); KEGG [24] (KEGG Ortholog database); and GO (Gene Ontology) [25]. We used the software of BLAST and an e-value of “1e-10” in the NT database analysis; the software of Diamond BLASTX and an e-value of “1e-10” in the NR, KOG, Swiss-Prot, and KEGG databases analysis; and the software Hmmscan in the Pfam database for family identification.

The *M. foliosa* transcription factors were identified using the animal transcription factors (TFs) database animalTFDB 2.0 [26]. HMMSearch was used to identify TFs according to the Pfam files of the transcription factor family. The *Symbiodinium* transcription factors were predicted using iTAK software [27].

Long non-coding RNA (LncRNA) are a class of RNA molecules whose transcripts exceed 200 nt in length and do not code for proteins. We used the Coding–Non-Coding Index (CNCI) [28], Coding Potential Calculator (CPC) [29], Pfam-scan [30], and PLEK [31] tools to predict the coding potential of the transcripts. We used the CNCI with default parameters. In CPC, the NCBI eukaryotes protein database was used and set the e-value to “1 × 10^−10^”. We translated each transcript in all three possible frames and used Pfam Scan to identify the occurrence of any of the known protein family domains documented in the Pfam database. Any transcript with a Pfam hit was excluded in the following steps. The Pfam searches used the default parameters of -E 0.001–domE 0.001. The PLEK SVM classifier used an optimized K-mer approach to construct the best classifier to assess the coding potential for species lacking high-quality genomic sequences and annotations. PLEK used the default parameters of -minlength 200. The transcripts not predicted with coding potential by all three of the tools above were filtered out, and those without coding potential were candidate sets of lncRNAs.

#### 2.4.4. Coral and Symbiodinium Gene Identification

We found the genomes of the genus Montipora in the NCBI genome database and then compared the transcript data with three *Montipora* species, including *Montipora capitata* [32], *Montipora efflorescens* [33], and *Montipora cactus* [33]. Simultaneously, the transcript data was compared with six zooxanthellae genomes, including *Symbiodinium kawagutii* [13], *Symbiodinium microadriaticum* [34], *Symbiodinium pilosum* [35], *Symbiodinium necroappetens* [35], *Symbiodinium natans* [35], and *Breviolum minutum* [36,37,38]. Finally, the transcriptome was divided into *M. foliosa* coral and symbiotic symbiodinium (Appendix A). We used the default parameters (-ax splice:hq -uf and -G 200k for PacBio Iso-seq) of the minimap2 (version 2.17) software for all alignments [39]. We set the mapping quality threshold to be greater than 10 to ensure the accuracy rate was greater than 0.9 for all alignments [40].

#### 2.4.5. Gene Expression Quantification

The CD-HIT software was used to remove redundancy in the corrected consensus sequences, and the obtained transcripts were used as the reference sequence (REF) of the genes. Then, clean reads of each sample obtained from Illumina sequencing were mapped back to the REF for quantification. Gene expression levels were estimated by RSEM [41] for each sample: (1) Clean data were mapped back onto the transcript sequences; (2) the read count for each transcript was obtained from the mapping results; and (3) the read count was transformed to fragments per kilobase million (FPKM) [42]. Bowtie2 of the comparison software in RSEM was set to enter an end-to-end and sensitive mode, and the other parameters were set to default. Because TPM (Transcripts Per Kilobase Million) is a better indicator than FPKM [42], we therefore converted FPKM to TPM here, according to the literature [42].

#### 2.4.6. GO, KEGG, KOG/COG, and Pfam Enrichment Analysis

We used Molecular Signatures DataBase (MSigDB) software [43] to test the statistical enrichment of the genes ranked in descending order in the KEGG and GO.

We divided the transcripts into different gene sets according to the GO, KEGG, KOG/COG, and Pfam classifications. Then, we performed a Fisher’s exact test on each class in the ordered genes or the user-specified gene list. A *p*-value less than 0.05 indicates significantly enriched terms for each database.

## 3. Results

### 3.1. Polymerase Read Statistics

The platform for the PacBio Sequel sequencing is circular sequencing. High-quality sequencing reads produced by a single molecule in the process of sequencing are known as polymerase reads. The statistical results of the polymerase reads are shown in Table 1.

### 3.2. Correction of Transcripts and Removal of Redundant Transcripts

Second-generation data were used to calculate the sequence length before and after the transcription correction, and the results are shown in Table A1.

According to the 95% similarity between the sequences, the corrected transcript sequences were clustered to remove redundancy, and the length-frequency distribution before and after the removal of the redundant transcripts was counted, as shown in Table A2 and Table A3 and Figure A1.

### 3.3. Gene Function Annotation

To obtain comprehensive gene function information, gene function annotation was carried out on the sequences. There were 19,773 genes annotated in at least one database in *M. foliosa* and 8893 genes annotated in all seven databases. There were 199 genes annotated in at least one database in zooxanthellae and 28 genes annotated in all seven databases. All the statistical results are shown in Figure 1 and Appendix A.

#### 3.3.1. NR Database Annotation Results

A total of 18,570 *M. foliosa* genes and 128 *Symbiodinium* genes were annotated in the NR database, and the statistical results are shown in Appendix A and Figure A1.

#### 3.3.2. GO Classification

After GO annotation was performed on the genes, 14,129 *M. foliosa* genes and 95 *Symbiodinium* genes were annotated (Appendix A), and the classification results are shown in Figure 2.

#### 3.3.3. KOG Classification

A total of 13,588 *M. foliosa* genes and 70 *Symbiodinium* genes were annotated in the KOG/COG database (Appendix A). The genes were classified according to the KOG/COG group, and the results are shown in Appendix A.

#### 3.3.4. KEGG Classification

A total of 17,859 *M. foliosa* genes and 124 *Symbiodinium* genes were annotated in the KEGG database (Appendix A). The genes were classified according to the KEGG category in which they participated, as shown in Figure 3.

#### 3.3.5. Pfam Database Annotations

A total of 14,129 *M. foliosa* genes and 95 *Symbiodinium* genes were annotated in the Pfam database (Appendix A).

#### 3.3.6. Swiss-Prot Database Annotations

A total of 15,672 *M. foliosa* genes and 106 *Symbiodinium* genes were annotated in the Swiss-Prot database (Appendix A).

### 3.4. Gene Structure Analysis

#### 3.4.1. CDS Prediction

A total of 20,230 *M. foliosa* transcripts and 129 *Symbiodinium* transcripts were predicted to have CDS regions (Appendix A), and the statistical results are shown in Figure 4.

#### 3.4.2. Transcription Factor Analysis

A total of 847 *M. foliosa* transcripts and 2 zooxanthellae transcripts (zf-MIZ, p53) were predicted as TFs (Appendix A). The length distribution of the TFs is shown in Figure 4.

#### 3.4.3. LncRNA Prediction

A total of 6328 transcripts are predicted to be lncRNAs, none of them belonging to zooxanthellae (Figure 5).

### 3.5. Gene Expression Analysis

The statistical results are shown in Figure 5 and Table A4. All the gene expression results are shown in Appendix A.

#### 3.5.1. Gene Enrichment Analysis of All Zooxanthellae

To study the function of the captured zooxanthellae genes, we performed enrichment analysis on the four databases (KEGG, GO, KOG, and Pfam, Figure 6). All of the enrichment analysis results are shown in Appendix A.

#### 3.5.2. Enrichment Analysis of M. foliosa Highly Expressed Genes

To study the functions and biological processes of genes with different expression levels, we ranked the expression values of the unique genes and performed gene set enrichment analysis (GSEA) based on the GO (Figure 7), KEGG, KOG, and Pfam (Appendix A).

#### 3.5.3. Enrichment Analysis of M. foliosa Metabolism-Related Genes Located on the Cell Membrane

The nutrient exchange between coral and zooxanthellae is carried out through proteins located on the membrane. Symbiotic zooxanthellae are produced by coral phagocytosis; so, the corals provide the outermost membrane of the endosymbiont cells. Therefore, we performed enrichment analysis on the metabolism-related genes located on the cell membrane in the *M. foliosa* transcriptome. Three pathways of Glycan biosynthesis were discovered: Mucin type O-glycan biosynthesis, other types of O-glycan biosynthesis, and Glycosphingolipid biosynthesis. Through GO molecular function studies, we found that these membrane proteins are highly expressed in five signaling pathways related to sugar metabolism, including Acetylglucosaminyltransferase activity, Acetylgalactosaminyltransferase activity, Galactosyltransferase activity, Transferase activity, transferring glycosyl groups, Transferase activity, and transferring hexosyl groups (Table 2).

## 4. Discussion

### 4.1. Gene Function of All Zooxanthellae Related to Symbiosis

The most significantly enriched KEGG pathways related to symbiosis include photosynthesis and nitrogen metabolism. Photosynthesis is the most important biological process in the symbiosis of corals and zooxanthellae. As corals live in relatively poor nutrition, the photosynthesis of zooxanthellae is the main source of organic matter in a symbiotic system [44]. Highly dependent on the availability of nitrogen in coral hosts, the nitrogen cycle may be critical to the stability of coral–algae symbiosis and overall biological function, especially when the nutrients are depleted. The interference of the nitrogen cycle may be closely related to coral bleaching and disease [45].

The molecular functions that are significantly enriched and related to symbiosis include kinase activity, heme binding, cysteine-type peptidase activity, and carbonate dehydratase activity. Rosic et al. explored the possible role of conserved calcium/calmodulin-dependent protein kinases (CCaMKs) and the inositol pathway for the foundation of cnidarian–dinoflagellate symbiosis [46]. Notably, after eight days of thermal challenge, these bleaching corals showed a marked increase in a heme-binding protein 2-like homolog [47]. A type of Cysteine-type protease (VLK protease or VLKP) belonging to the peptidase C1A superfamily exhibited maximum activity during the late logarithmic growth phase; these attributes suggest that VLKP is involved in the metabolism of proteins in acidic organelles [48]. Carbonic anhydrase activity was found in 28 of the 29 species of seaweeds and angiosperms collected near Lizard Island in Australia’s Great Barrier Reef, as well as mixed cultures of phytoplankton. Host tissues contain approximately five times the activity of their respective zooxanthellae. It is impossible to infer a correlation between hypercarbonic dehydratase activity and calcium carbonate deposition [49].

Photosynthesis of the symbiont plays a central role in coral–algae symbiosis because most of the host’s metabolic needs come from the photosynthetic fixation of carbon. The carbon concentration mechanism (CCM) is used to enhance the photosynthesis of symbionts, in which carbonic anhydrase (CA) plays an important role in the accumulation, transportation, and mutual transformation of inorganic carbon [4].

### 4.2. Functions of Highly Expressed M. foliosa Genes Related to Symbiosis

#### 4.2.1. GO Entries Related to Symbiosis

Subcellular localization. The genes with the highest expression levels are located intracellularly, in the ribosome, gas vesicle, and nucleosome.

Cell–matrix adhesion. Research shows that scleractinian coral *Stylophora pistillata* gastrodermal cells share the expression of adhesion proteins [50]. Calicoblastic cells are also enclosed with mesoglea, a sheet of extracellular matrix, and attached to the coral skeleton by desmocytes via a hemidesmosome adhesion complex. Maintaining a relatively stable internal environment for corals is critical for both symbiosis and calcification [4].

Bioluminescence. Many deep sea corals can produce bioluminescence [51], but the biological significance of coral luminescence is not yet known. Some research believes that luminescence can attract zooxanthellae and help corals recover from bleaching [52].

Cellular iron ion homeostasis. As a stress protein, iron storage proteins are transcriptionally upregulated by corals during thermal bleaching, possibly in response to the release of iron caused by ROS disrupting the heme–ferritin binding [53,54]. GO term ferric iron binding and 2 iron = 2 sulfur cluster binding were also significantly enriched, which suggests that iron metabolism may have an important role in coral stress.

Glutamine biosynthetic process. As a key nutrition exchange molecule, glutamine can directly transfer from a dinoflagellate to a coral host [55].

#### 4.2.2. KEGG, KOG, and Pfam Entries Related to Symbiosis

KEGG entries with significant enrichment involve neurodegenerative diseases and metabolism (oxidative phosphorylation and nitrogen metabolism). In calicoblastic cells, ATP supply may contribute to light-enhanced calcification in corals more than abiotic mechanisms [56], and oxidative phosphorylation is the main source of ATP.

KOG protein function analysis found that *M. foliosa* overexpressed ribosomal-related genes and ferritin in inorganic ion transport and metabolism. Ferritin is an indispensable intracellular protein that acts as a buffer to maintain the balance of iron. The intracellular iron store can prevent the spread of infective agents by impeding their proliferation [57]. We also found that in the Pfam database enriched items, the ferritin-like domain was enriched in coral. The question of how invading algae manage to avoid intracellular attacks in host phagocytes has been of interest for decades. Fitt and Trench [58] found that both ferritin and acid phosphatase colocalize to symbiosomes with heat-killed ingested *S. microadriaticum*, while symbiosomes containing healthy dinoflagellates remain free of either marker.

In Pfam enrichment, the green fluorescent protein (GFP) protein family is significantly enriched in the Pfam database. Aihara et al. [59] demonstrated that corals can attract free-living *Symbiodinium* in the environment by endogenous GFP-related green fluorescence.

### 4.3. Molecular Biological Mechanisms of Intracellular Symbiosis

The symbiosis of coral–zooxanthellae is essential for corals. Two of the key questions for coral symbiosis research are recognition of symbiotic algae and host cells and the exchange of nutrients between them (Figure 8).

Not only corals, but also cnidarian can form a cellular symbiosis relationship with dinoflagellates. Compared with corals, the sea anemone *Exaiptasia pallida* (or *Aiptasia*), which is closely related to corals [3], and green *Hydra* [4] are model organisms for studying coral–zooxanthellae symbiosis. Compared with corals, sea anemones and green *Hydra* are easier to cultivate under laboratory conditions, and many of the molecular mechanisms of symbiosis in coral cells are derived from these model organisms. Cnidaria-Dinoflagellate symbiosis involves a series of stages. The first stage is the contact between cnidarians and symbiotic algae; they recognize each other through ligands and receptors on the cell surface. In the second stage, the symbiotic algae are swallowed by the cnidarian, producing phagocytic vesicles. The third stage is intracellular sorting. Incompatible symbiotic algae and destroyed cells are degraded by host cells, leaving only algae that can symbiotically form a stable symbiotic body with the host. The fourth stage is the dynamic equilibrium stage, where the cell proliferation between the host and the symbiotic algae and the exchange of nutrients reach a symbiotic balance. The fifth stage is the symbiosis imbalance stage. The symbiotic algae are degraded or eliminated from the host cell, or the host cell dies [3,4].

### 4.4. Symbiosis Recognition-Related Genes

The first step for symbiosis is to recognize Symbiodiniaceae by coral partners; this process involves a variety of molecules located in the cell surface or secreted by host cells called pattern recognition receptors (PRRs). PRRs can recognize signature microbial compounds named microbe-associated molecular patterns (MAMPs) [3,4,60]. PRRs include the complement 3 receptor, thrombospondin type I repeats, sphingosine rheostat, transforming growth factor beta, nucleotide-binding oligomerization domain proteins, scavenger receptors, and toll-like receptors. The number of transcripts of these genes, the average isoform counts, and the average expression value are shown in Table 3 (see Appendix A for all key genes related to recognition).

The complement system originates in the deuterostomes [61] and is an important component of the innate immune system. It contains at least 20 proteins in humans that respond to the recognition of the molecular components of microorganisms and are sequentially activated in an enzyme cascade—the activation of one protein enzymatically cleaves and activates the next protein in the cascade [62,63]. According to their biological functions, they can be divided into three categories: 1. Inherent components of the complement, which exist in plasma and body fluids and participate in various molecules of complement activation, including various protein molecules in the three activation pathways and their downstream common components, such as C1 to C9. 2. Complement regulatory protein, a protein molecule that exists in the plasma and on the surface of cell membranes and controls the intensity and range of complement activation by regulating key enzymes in the pathway of complement activation. 3. Complement receptors (CRs), which exist on the cell membrane and can combine with active fragments formed after complement activation and receive receptor molecules with a variety of biological effects. The components of the complement system are all glycoproteins [64,65]. There are three pathways of complement activation in the human: the classical pathway, the alternative pathway, and the lectin pathway. The main activators of the classical pathway are antibodies such as IgG and IgM, which play a role in the later stages of infection. Activators of the alternative pathway include bacterial, fungal, or viral infections. The activator of the lectin pathway is mainly N-galactosamine or mannose on the surface of microorganisms, which is recognized by mannose-binding lectin (MBL) or fibrocollagen (ficolin FCN). The lectin pathway plays an important role in the immunity of non-deuterostome invertebrates [61]. The final common target of the complement system is to assemble MAC (membrane attack complex) on the cell membrane and mediate the cytolysis. At the same time, a variety of cleavage fragments produced during complement activation can interact with the cell membrane receptors, such as type I complement receptors (CR1, C3b/C4b receptors), type II complement receptors (CR2, C3b receptors, CD21), type III complement receptor (CR3), and type IV complement receptors (CR4), C5aR, C3aR, C1qR, to mediate a variety of biological functions. Some factors (such as Factor B) in the alternative pathway, mannose-binding lectin associated serine proteases (MASP) in the lectin pathway, and homologues of C3, a key molecule in the complement system, have been found in invertebrates [62]. Studies have found that the homologous genes of the components of the vertebrate complement system play a key role in cnidarian–dinoflagellate symbiosis at the recognition stage and in the regulation of cnidarian–dinoflagellate symbiosis [16,66]. We found that the *Acropora digitifera* homologous gene complement C2-like, complement C3, and component C1q subcomponent-binding gene exist in *M. foliosa* (Appendix A). Among them, C1q participates in the recognition of antibody molecules in the human body and can activate C2. C2 can activate C3, and C3 and its downstream molecules are shared by the three pathways.

Lectins are one of the activation pathways of the complement system and are responsible for identifying pathogens invading the cell [67]. As the main component of PPRs, lectin protein plays a key role in the identification of cnidarian-dinoflagellate [3]. Zooxanthellae interact with lectin molecules on the surface of cnidarian cells through sugar residues (such as mannose) and finally initiate phagocytosis to form phagosomes. We found multiple *M. foliosa* homologous genes belonging to the lectin family. In *A. digitifera*, C-type lectin, galactoside-binding lectin, and mannose lectin ERGIC-53 are involved in glycoprotein traffic, etc. Homologs of C-type lectin and jacalin-like, lectin domain-containing, protein-coding genes were found in *E. pallida*. These genes will provide important references for studying the symbiotic relationship between coral and zooxanthellae (Appendix A).

Thrombospondin’s proteins are multidomain glycoproteins that function on the cell surface and in the extracellular matrix environment. The thrombospondin type I repeat (TSR) superfamily member protein is widely present in basal metazoans [68]. The TSR domain consists of approximately 60 amino acids and has several highly conserved motifs [69]. The TSR protein superfamily is widespread in eukaryotic cells [69,70]. The TSR superfamily members are diverse, which indicates that the highly conserved TSR domain superfamily members have been replicated and shuffled many times. The role of Thrombospondins was first characterized in mammals. They are extracellular, multi-domain, calcium-binding glycoproteins that play a pleiotropic tissue-specific role and involve interactions with cell surfaces, cytokines, and the extracellular matrix [69]. In mice, TSR protein has been shown to interact with a variety of proteins to regulate cell proliferation, migration, and apoptosis [71]. The function of TSR in cnidarians mainly refers to vertebrates. TSR is similar in sequence to the vertebrate complement protein properdin. In vertebrates, the three functions of properdin that may be related to symbiosis are: 1. Properdin can act as a member of pattern recognition receptor (PRR) to recognize MAMPs. 2. As a member of an alternative pathway of the complement system, it participates in the activation and stabilization of C3 [72]. 3. It participates in the transforming growth factor (TGFb) pathway. TSR proteins have been identified in several cnidarian species that are differentially expressed in the symbiotic state [68,73]. There is increasing evidence that cnidarians possess many genes containing TSR domains. For example, it is considered to be related to microbe binding in *Hydractinia symbiolongicarpus* [68,74]. At the beginning of Aiptasia’s symbiosis, blocking TSR with anti-TSR antibodies can reduce the colonization rate, while the addition of exogenous TSR peptides causes the colonization success rate to exceed the control level [3]. In *A. digitifera*, we found the homologous gene disintegrin metalloproteinases with thrombospondin repeats and the Serine proteinase inhibitor (KU family) with thrombospondin repeats; in *Nematostella vectensis* and *E. pallida*, we found the homologous gene disintegrin metalloproteinases with thrombospondin repeats. These are genes belonging to the thrombospondin superfamily (Appendix A).

TGF-β belongs to the transforming growth factor superfamily. The roles of TGF family members in vertebrates include developmental control, cell proliferation control, and immunity. The homologs corresponding to TGF-β (which in the following will be referred to as TGF-β) in vertebrates are an important component in the recognition stage of coral–zooxanthellae symbiosis in coral. It generates immunosuppressive signals by interacting with SR and TSR [3]. Some research has shown that TGF-β is highly expressed in the early stage of symbiosis [75]. Another study [76] showed that mixing excess TGF-β with coral under high-temperature pressure alleviates the bleaching process. The role of TGF-β in cnidarian–dinoflagellate symbiosis is supposedly immunosuppressive. In *A. digitifera*, we found the transforming growth factor-beta receptor-associated protein 1-like gene, the latent-transforming growth factor beta-binding protein 1-like gene, the AdiTgf2 gene for transforming growth factor 2 protein, the TGFbeta receptor signaling protein SMAD, and related protein genes. In *E. pallida*, we found the homologous gene of *M. foliosa* transforming growth factor beta regulator 1, the TGFbeta receptor signaling protein SMAD and related proteins, translation initiation factor 3, and subunit i (eIF-3i)/TGF-beta receptor-interacting protein (TRIP-1). These genes belong to the transforming growth factor beta superfamily.

Scavenger receptors (SRs) are a large and diverse superfamily of cell surface receptors. Scavenger receptors in vertebrates are known to be involved in a wide range of processes, such as homeostasis, apoptosis, inflammatory diseases, and pathogen clearance. The scavenger receptor superfamily is able to recognize and bind a broad range of common ligands. These ligands include polyanionic ligands, such as lipoproteins, apoptotic cells, cholesterol ester, phospholipids, proteoglycans, ferritin, and carbohydrates. Many research works have proven that SR homologs in cnidarian are involved in cnidarian–dinoflagellate interactions in the recognition and phagocytosis stages [3]. In our research, 15 genes were identified as being of the latent scavenger receptor family as they are annotated to the scavenger receptor of the nearest species, *A. digitifera* and *E. pallida*. Several genes in *M. foliosa* are annotated to the scavenger receptors family. In *A. digitifera*, we found glutamate decarboxylase/sphingosine phosphate lyase, macrophage scavenger receptor types I and II-like isoform X2, and scavenger receptor cysteine-rich type 1 protein M160-like. The scavenger receptor cysteine-rich type 1 protein M130, a homologous gene of *M. foliosa*, was found in *E. pallida* (Appendix A).

Sphingolipids are amphoteric lipids that contain a sphingolipid skeleton, and they are an important component of plasma membranes. Sphingolipids and their metabolites are important molecules involved in many important signal transduction processes, such as the regulation of cell growth, differentiation, senescence, and programmed cell death [75]. The sphingosine rheostat, which includes the sphingosine (Sph), formed by the action of sphingosine-1-phosphate phosphatase (SGPP), and sphingosine 1-phosphate (S1P), formed by sphingosine kinase (SPHK), is a key homeostatic regulatory pathway known to function in cell fate and immunity in animals [77]. The sphingosine rheostat (including SGPP and SPHK) is known as a regulator of the homeostatic of Sph and S1P. The high expression of rheostat was first found in *Anthopleura elegantissima* [78]. Compared to nonsymbiotic controls, symbiotic animals show higher levels of pro-survival sphosphinol-1 phosphate (S1P) and higher amounts of the enzymes that drive S1P production [3,79]. An excess of S1P under high-temperature pressure alleviates the bleaching process in cnidarian–dinoflagellate symbiosis [80]. We found an *A. digitifera* homologous gene in *M. foliosa*: Glutamate decarboxylase/sphingosine phosphate lyase (Appendix A).

Nucleotide-binding and Oligomerization Domain 2 Protein (NOD2) is a member of the NOD 1/APAF-1 family, which encodes a protein containing two caspase (CAD) domains and six leucine-rich repeat sequences (LRRs) [81]. It plays an important role in the immune response of intracellular bacterial lipopolysaccharide (LPS) and participates in the immune response of intracellular bacterial lipopolysaccharide (LPS) by recognizing intracellular bacterial lipopolysaccharide (LPS)-derived Muramyl dipeptide (MDP) and activating intracellular NF-κB protein [81]. NODs are a component of the PRRs involved in coral–zooxanthellae symbiosis [4]. In our research, nine transcripts annotated as human and mouse NOD homologs were identified. We found that seven transcripts can be annotated to the *Mus musculus* nod2 gene. Two transcripts can be annotated to the *Homo sapiens* nod1 gene. No corresponding homologous genes were found in scleractinia and cnidarians.

Homology analysis found that TLR originated from the cnidaria phylum and participated in innate immunity in cnidarians. Toll-like receptors (TLRs) belong to the inherent immune pathogen pattern recognition receptors, which can identify the proteins, nucleic acids, and lipids of pathogenic microorganisms that invade the body in humans, as well as the intermediate products and metabolites synthesized in the reaction process. Toll-like receptors activate inflammatory signaling pathways such as NF-κB. NF-κB is found in almost all animal cells and is involved in the response of cells to external stimuli, such as cytokines, radiation, heavy metals, and viruses. NF-κB plays a key role in the cellular inflammatory and immune responses. The NF-κB gene sequence is highly conserved in animals. Homologs of the TLR in the TLR-to-NF-κB pathway are found in *N. vectensis*, *A. digitifera*, Hydra, *Orbicella faveolata*, and *E. pallida* [82]. In *Nematodes*, *Drosophila*, and mammals, TLR is involved in many biological processes, such as innate immunity and development. Some evidence suggests that invertebrate (such as cnidarians) TLR/LRR proteins can detect bacterial flagellin, and some mollusk TLR proteins can detect multiple ligands. Studies on cnidarians, mollusks, and annelids have shown that pathogen treatment increases the expression of genes in the TLR-to-NF-κB pathway, which indicates that lower animals respond to infection through the upregulation of innate immune pathways. Most cnidarians have an almost complete innate immune signaling pathway [83,84]. Studies have shown that the presence of symbiont in AIPTASIA can trigger significant immunosuppression [80,85]. Studies have shown that adding zooxanthellae to larvae that have not yet symbiotically with symbionts reduces the expression of NF-κB [73,86]. In contrast, the expression is increased during stress-induced adult bleaching [86], leading to a result similar to that of heat-stress functional genomics studies conducted in coral *Acropora palmata* [87]. These research works indicate that repressing NF-κB can inhibit the host immune response and hence improve the success rate of symbiosis. In our research, 30 genes were identified as belonging to the toll-like receptor family; the homolog predicted as *Mus musculus* toll-like receptor 6 (TLR6) has two isoforms, while the others have only one isoform. In *A. digitifera*, we found the homologous genes of *M. foliosa* toll-like receptor 1, toll-like receptor 2, toll-like receptor 6 isoform X3, toll-like receptor 6 isoform X5, and toll-like receptor 6.

Rab protein is a small GTP binding protein, which is highly conserved in organisms. Through its interaction with upstream and downstream proteins, Rab plays an important role in different stages of vesicle transport. The study of the sea anemone *Aiptasia pulchella* found that the cnidarian orthologs of human Rab5 and Rab7 were closely related to the maturation or arrest of phagocytic vesicles [60]. Rab5 appeared around healthy new intakes and established Symbiodinium (arrest), but not around heat-inactivated or DCMU-treated newly ingested Symbiodinium (maturation). In contrast, anti-human Rab7 is located around newly ingested symbiotic bacteria that are heat-inactivated or treated with DCMU, but does not exist in untreated new infections or established symbiotic bacteria [4,60]. Similarly, symbionts containing healthy Symbiodinium are Rab11 negative and Rab4 positive. It was found that Rab4 was immediately recruited into all early phagosomes but only retained in those phagosomes containing healthy symbionts, indicating that Rab4 is essential for the production of symbisome [88]. Rab11 is rapidly recruited in the initial stage of phagosome formation, but is excreted from phagosome before the stable stage of the endosymbiotic relationship [88]. We discovered several *M. foliosa* homologous genes in the Rab family. In *A. digitifera*, we found the *M. foliosa* homologous genes Rab1, Rab2, Rab3, Rab4, Rab5, Rab11, Rab18, Rab21, Rab27, and Rab39. *M. foliosa* homologous genes Rab1, Rab3, Rab5, and Rab6 were found in *N. vectensis* and *E. pallida*.

### 4.5. M. foliosa May Convert the Monosaccharides Transported by Zooxanthellae into Glycans, Thereby Avoiding the Fluctuation of the Sugar Concentration in the Cell

In the stable phase of cnidarian-dinoflagellate symbiosis, continuous material exchange between them is required. At present, we still lack a clear understanding of the specific form of material exchange. Some studies believe that carbohydrates are not only an important component to identify and maintain symbiosis [4], but also the nutrients of zooxanthellae which may also be transferred to coral cells in the form of monosaccharides [4,15]. The cell membrane is the only way for material exchange. Through the analysis of 3.5.3, we found that these metabolic-related membrane proteins are significantly enriched in functions related to glycan biosynthesis. Therefore, we speculate that in the process of sugar transfer to coral cells, it is first converted into glycogen molecules on the cells, thereby avoiding the influence of zooxanthellae photosynthesis on the concentration of sugar in coral cells (Table 2).

In summary, first, we obtained high-quality *M. foliosa* transcripts through the Pacbio and illumina platforms and annotated their functions. Second, we identified different isoforms of *M. foliosa* transcripts. Third, we performed enrichment analysis on the highly expressed *M. foliosa* gene and all the *Symbiodinium* genes for GO, KEGG, Pfam, and KOG databases. We discussed functional enrichment annotations related to symbiosis. Fourth, we summarized the genes related to coral–zooxanthellae symbiosis and discussed the genes related to symbiosis. Our work provides valuable transcriptome data for the study of the molecular mechanism of coral–zooxanthellae symbiosis.

## 5. Conclusions

The coral–zooxanthellae symbiosis system plays a vital role in maintaining the health of the coral reef ecosystem. Studying the mechanism of coral–zooxanthellae symbiosis involves multiple processes, such as symbiont recognition and maintenance. Due to the relatively short read length of the second-generation genome sequencing, on the other hand, due to the lack of effective reference genome data in corals, the current assembly accuracy of the *M. foliosa* transcriptome is relatively low, and the isoform of different transcripts cannot be identified. In this article, for the first time, the transcriptome of *M. foliosa* of reef-building corals was sequenced for third- and second-generation sequencing technology. Additionally, the symbiote recognition, membrane protein, and metabolism-related genes in the coral–zooxanthellae symbiotic system were summarized. At the same time, GO and KEGG were enriched for highly expressed genes, providing relatively reliable transcriptome sequence data for future research. Next, we will study the dynamic process of symbiont escape under stress conditions (e.g., ocean acidification and heat stress), such as changes in gene expression at different times.

## Figures and Tables

**Figure 1 biology-10-01274-f001:**
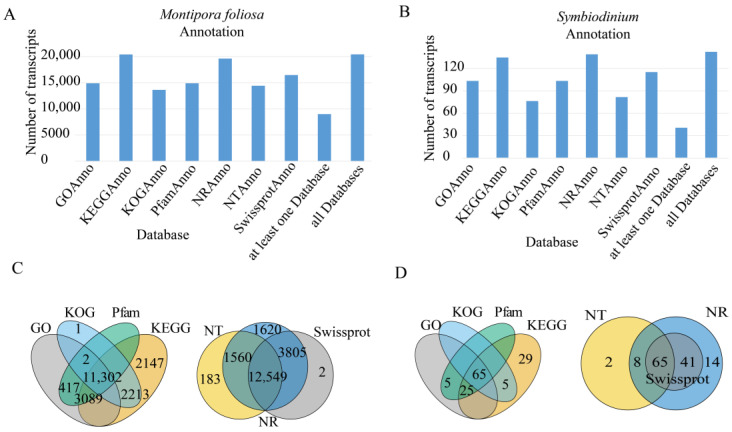
Annotated statistical results of seven major databases. Pfam (protein family), KOG/COG (Clusters of Orthologous Groups of proteins), KEGG (KEGG Ortholog database), and GO (Gene Ontology) are function annotation databases. NR (NCBI non-redundant protein sequences), NT (NCBI non-redundant nucleotide sequences), and Swiss-Prot (a manually annotated and reviewed protein sequence database) are sequence annotation databases. At least one database: the number of annotated transcripts in at least one database; all databases: number of transcripts annotated in all databases. (**A**) *Montipora foliosa* unique gene annotation by the seven databases. (**B**) *Symbiodinium* unique gene annotation by the seven databases. (**C**) The overlaps of *Montipora foliosa* gene annotation among the seven databases. (**D**) The overlaps of *Symbiodinium* unique gene annotation among the seven databases. Each large circle represents a database, and the sum of counts in the circle represents the total number of transcripts that can be annotated by this database, while the number in the overlapping portion of the circle represents the transcript annotation results shared between the databases.

**Figure 2 biology-10-01274-f002:**
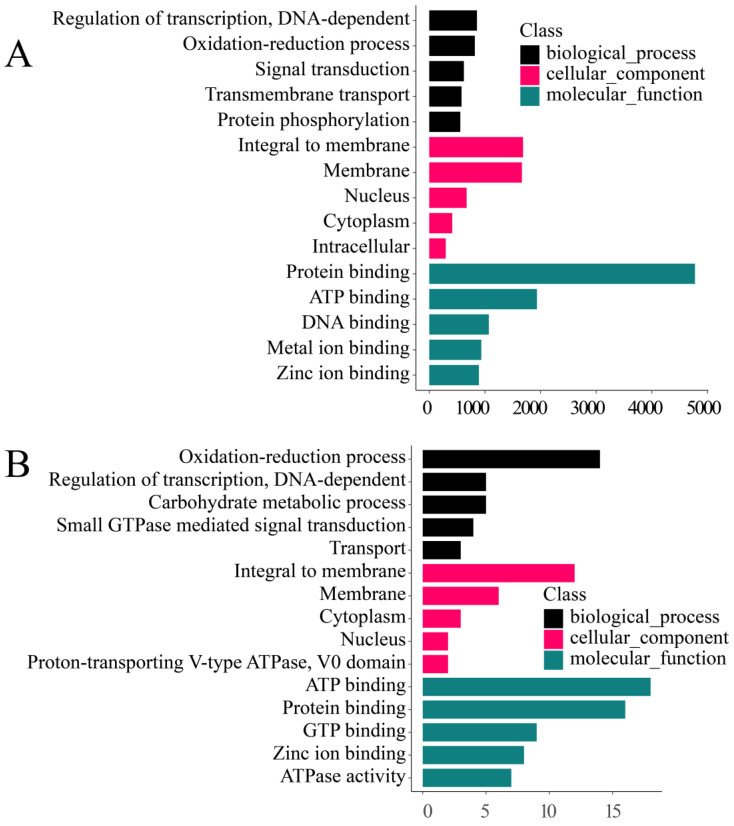
GO (Gene ontology) statistics. Each scale on the ordinate represents a GO entry, and the abscissa represents the number of genes under the GO entry. The color represents GO classification. (**A**) *Montipora foliosa* GO annotation; (**B**) *Symbiodinium* GO annotation.

**Figure 3 biology-10-01274-f003:**
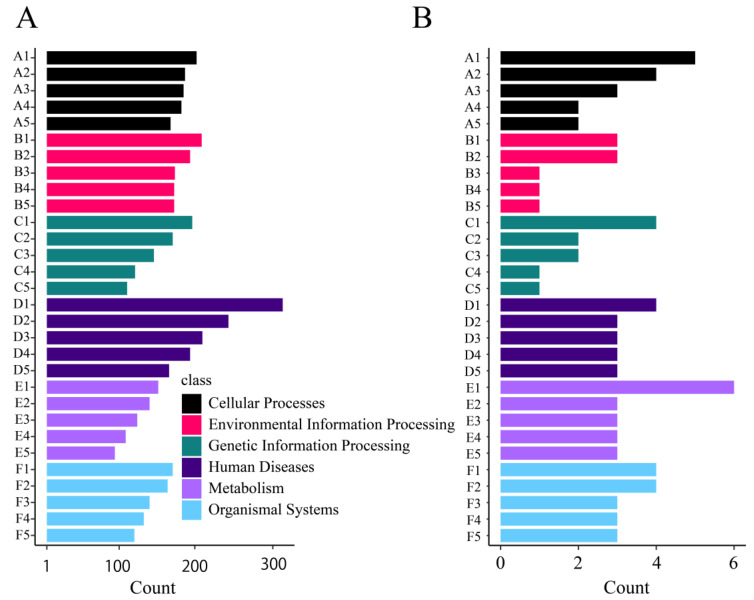
KEGG (KEGG Ortholog database) statistics. Each scale on the ordinate represents a KEGG entry, and the abscissa represents the number of genes under the KEGG entry. The color represents KEGG classification. (**A**) *Montipora foliosa* KEGG annotation; A1: Phagosome; A2: Gap junction; A3: Apoptosis; A4: Tight junction; A5: Meiosis-yeast; B1: AMPK signaling pathway; B2: PI3K-Akt signaling pathway; B3: MAPK signaling pathway; B4: ABC transporters; B5: cGMP-PKG signaling pathway; C1: Protein processing in the endoplasmic reticulum; C2: RNA transport; C3: Spliceosome; C4: Ubiquitin mediated proteolysis; C5: Protein processing in the endoplasmic reticulum; D1: Parkinson’s disease; D2: Pathways in cancer; D3: Prostate cancer; D4: Pathogenic Escherichia coli infection; D5: Alzheimer’s disease; E1: Nitrogen metabolism; E2: Purine metabolism; E3: Carbon metabolism; E4: Oxidative phosphorylation; E5: Carbon fixation in photosynthetic organisms; F1: Estrogen signaling pathway; F2: Antigen processing and presentation; F3: Longevity regulating pathway-multiple species; F4: Th17 cell differentiation; F5: Cardiac muscle contraction. (**B**) *Symbiodinium* KEGG annotation. A1: Endocytosis; A2: Focal adhesion; A3: Tight junction; A4: Adherens junction; A5: Regulation of actin cytoskeleton; B1: PI3K-Akt signaling pathway; B2: Rap1 signaling pathway; B3: cAMP signaling pathway; B4: MAPK signaling pathway; B5: Ras signaling pathway; C1: RNA transport; C2: Spliceosome; C3: Protein processing in the endoplasmic reticulum; C4: Ubiquitin mediated proteolysis; C5: Ribosome; D1: Pathways in cancer; D2: Proteoglycans in cancer; D3: MicroRNAs in cancer; D4: Epstein-Barr virus infection; D5: Breast cancer; E1: Carbon metabolism; E2: Purine metabolism; E3: Oxidative phosphorylation; E4: Biosynthesis of amino acids; E5: Fatty acid metabolism; F1: Insulin signaling pathway; F2: Thyroid hormone signaling pathway; F3: Adrenergic signaling in cardiomyocytes; F4: Oxytocin signaling pathway; F5: Neurotrophin signaling pathway.

**Figure 4 biology-10-01274-f004:**
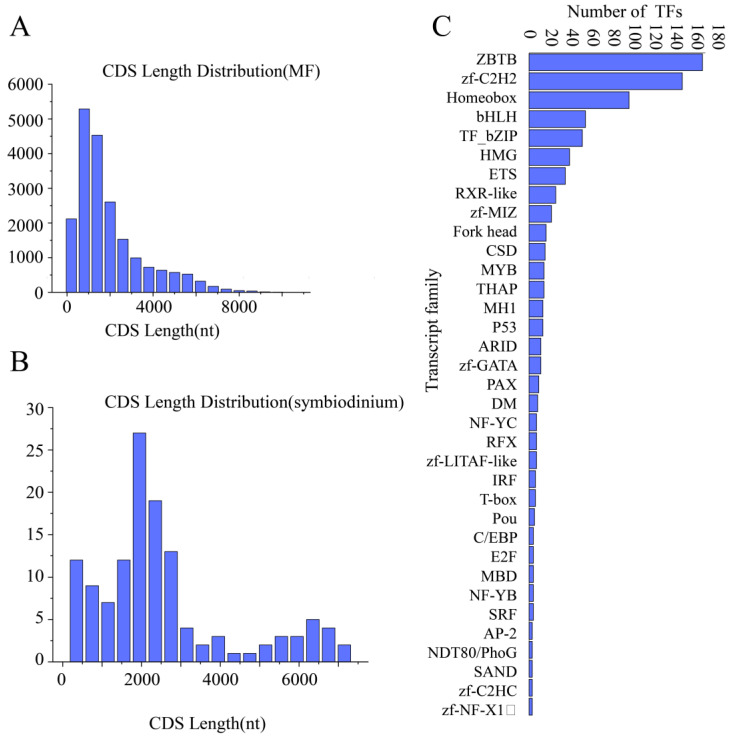
Statistical results of CDS (coding sequences) and TFs (Transcription factors). (**A**) The length distribution of the CDS of *Montipora foliosa* genes. (**B**) The length distribution of the CDS of zooxanthellae genes. (**C**) The distribution of the transcription factor family of *Montipora foliosa*.

**Figure 5 biology-10-01274-f005:**
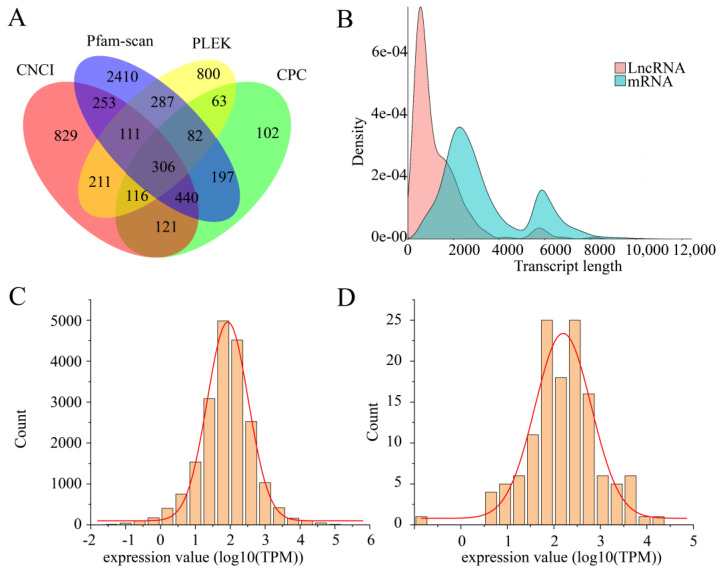
LncRNA and gene expression distribution. (**A**) Venn diagram of noncoding genes predicted by various software (CNCI, Pfam-scan, PLEK, and CPC). The sum of the number of each large circle represents the total number of lncRNA (long non-coding RNA) predicted by the software. The intersections of different circles represent the number of lncRNAs they share. (**B**) Comparative analysis of the length of lncRNA and mRNA. (**C**) The distribution of *Montipora foliosa* gene expression values. The abscissa represents the logarithm of the gene expression value with 10 as the base. The ordinate represents the number of genes. (**D**) Distribution of the gene expression values of zooxanthellae. The coordinate meaning is the same as (**C**).

**Figure 6 biology-10-01274-f006:**
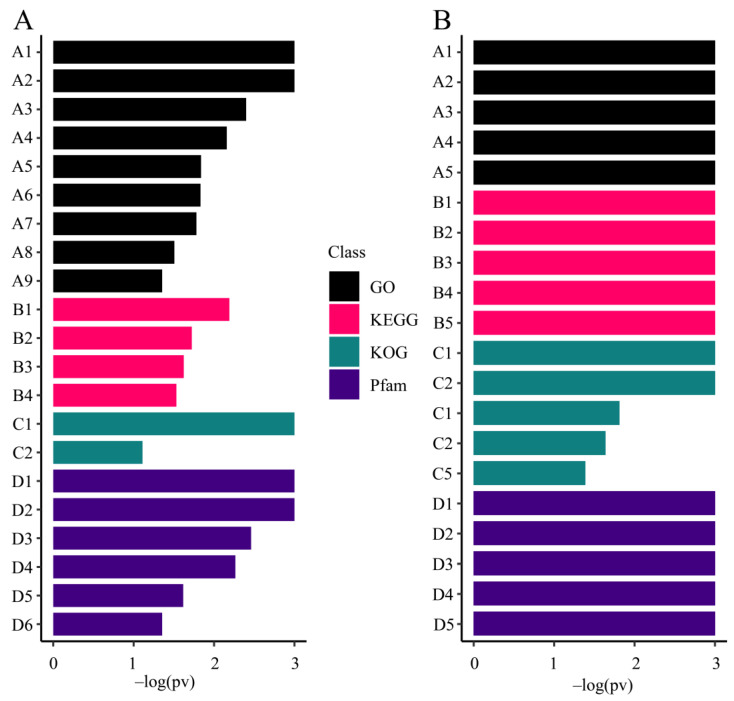
Enrichment analysis results of all zooxanthellae genes. For display convenience, all values of –log(pv) greater than 3 will be truncated to 3. (**A**) and *Montipora foliosa* special genes (**B**). The ordinate represents the significantly enriched entries in GO (Gene Ontology), Pfam (protein family), KOG/COG (Clusters of Orthologous Groups of proteins), and KEGG (KEGG Ortholog database). (**A**) A1: carbonate dehydratase activity; A2: oxidoreductase activity, acting on the CH-CH group of donors; A3: heme binding; A4: NAD binding; A5: kinase activity; A6: oxidation-reduction process; A7: iron ion binding; A8: carbohydrate metabolic process; A9: RNA helicase activity; B1: Vasopressin-regulated water reabsorption; B2: Starch and sucrose metabolism; B3: Salmonella infection; B4: Apoptosis; C1: Predicted carbonic anhydrase involved in protection against oxidative damage; C2: Dyneins, heavy chain; D1: Carbonic anhydrase; D2: Elongation factor Tu domain 2; D3: EGF-like domain; D4: ATPase domain predominantly from Archaea; D5: AAA domain (dynein-related subfamily); D6: RNA helicase. (**B**) A1: dynein complex binding; A2: chitin metabolic process; A3: motor activity; A4: transcription factor binding; A5: carbonate dehydratase activity; B1: Glutamatergic synapse; B2: Pathways in cancer; B3: Amphetamine addiction; B4: Dopaminergic synapse; B5: Insulin signaling pathway; C1: Fibroblast/platelet-derived growth factor receptor and related receptor tyrosine kinases; C2: Predicted carbonic anhydrase involved in protection against oxidative damage; C3: Tetraspanin family integral membrane protein; C4: Junctional membrane complex protein Junctophilin and related MORN repeat proteins; C5: Cysteine proteinase Cathepsin F; D1: Chitin binding Peritrophin-A domain; D2: Ankyrin repeat; D3: Integrase core domain; D4: Myosin tail; D5: IQ calmodulin-binding motif.

**Figure 7 biology-10-01274-f007:**
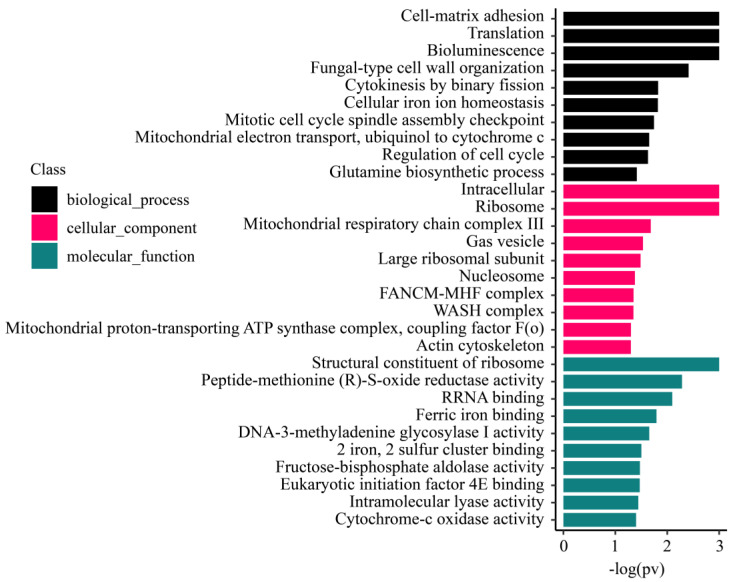
*Montipora foliosa* GO (Gene Ontology) gene enrichment analysis results. For display convenience, all values of –log(pv) greater than 3 will be truncated to 3. GO divided into biological processes, cellular components, and molecular functions, which are represented by different colors. The ordinate represents the significantly enriched entries in GO.

**Figure 8 biology-10-01274-f008:**
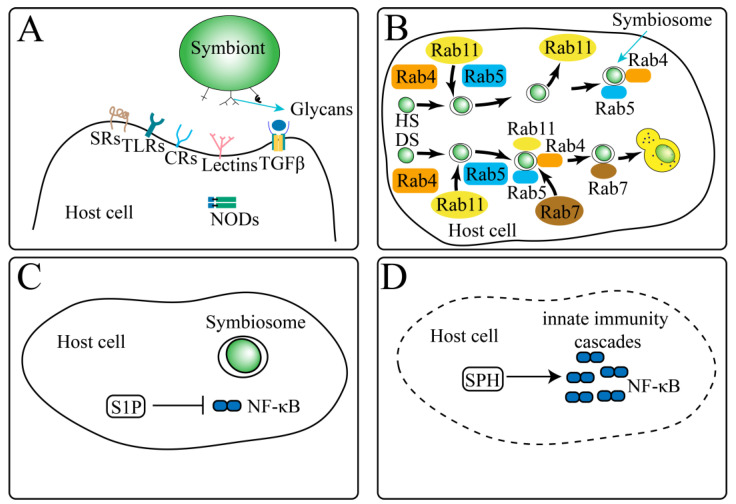
The molecular mechanism related to coral symbiosis. (**A**) Microbe-associated molecular pattern (MAMP in symbiont cell surface), pattern recognition receptors (PRRs in host cell surface), and interactions related to symbiont recognition. Glycan is mainly MAMP on the surface of symbiotic algae cells, and the relatively clear one is the mannose residue. PRR molecules mainly include scavenger receptors (SRs); thrombospondin type I repeats (TSR); toll-like receptors (TLRs); nucleotide-binding oligomerization domain proteins (NODs); Lectins; and TGFβ. (**B**) The sorting process of symbiotic algae in host cells. HS: Healthy Symbiont. DS: Damaged Symbiont. Among them, Rab4, Rab5, Rab7, and Rab11 are all homologous proteins of the corresponding proteins in higher vertebrate animals. Immediately after the symbiont enters the host cell, Rab4, Rab5, and Rab11 are combined with the symbiont and HS will dissociate Rab11 in the subsequent process to form a stable symbiosome. In the subsequent stage of DS, the symbiont will combine with Rab7 and Rab4, Rab5, and Rab11 dissociate. Then, it fuses with lysosomes and is digested. The sphingosine rheostat, which includes the sphingolipids sphingosine (Sph), is formed by the action of Sphingosine-1-phosphate phosphatases (SGPP), and sphingosine 1-phosphate (S1P), formed by sphingosine kinase (SPHK). It is a key molecule that regulates the immune mechanism of host cells. (**C**) In normal symbiotic cells, S1P occupies a higher proportion, and the NF-κB pathway is inhibited. (**D**) Under non-symbiotic infections or environmental stress conditions, Sph occupies a higher proportion, the NF-κB pathway is activated, and ultimately leads to host cell apoptosis.

**Table 1 biology-10-01274-t001:** Statistics of polymerase read by read bases, counts, and length.

Metrics	Value
Polymerase read bases (G)	43.11
Polymerase reads	699,225
Polymerase read length (mean)	61,652
Polymerase read N50	106,108

(1) Polymerase reads bases (G): data size of polymerase read. G: Giga bases (1,000,000,000 bp). (2) Polymerase reads: the number of polymerase reads; (3) Polymerase read length (mean): the average length of the polymerase reads; (4) Polymerase read N50: given a set of contigs, N50 is defined as the sequence length of the shortest contig at 50% of the total genome length.

**Table 2 biology-10-01274-t002:** Enrichment analysis of metabolism-related membrane protein genes.

Enrichment Class	Description	*p*-Value
KEGG	Mucin type O-glycan biosynthesis	0
KEGG	Pyrimidine metabolism	0.029366305
KEGG	Other types of O-glycan biosynthesis	0.03529412
KEGG	Glycosphingolipid biosynthesis	0.046674445
GO-MF	Acetylglucosaminyltransferase activity	0
GO-MF	Acetylgalactosaminyltransferase activity	0.001059322
GO-MF	Galactosyltransferase activity	0.001064963
GO-MF	Transferase activity, transferring glycosyl groups	0.002166847
GO-MF	Transferase activity, transferring hexosyl groups	0.045801528

GO-MF: gene ontology-molecular function; KEGG: Kyoto Encyclopedia of Genes and Genomes. All P-values less than 10 × 10^−9^ are set to 0.

**Table 3 biology-10-01274-t003:** Statistics of symbiosis recognition-related genes.

Components	Number of Transcriptions	Average Expression (TPM)	Average Number of Isoforms
Complement and its receptor	4	20.5	6
Lectin	70	24.32	1.54
Rab	84	30.7	1.27
Scavenger receptors	14	17.75	1.14
Sphingosine rheostat	4	15.67	1.5
Thrombospondin type I repeats	23	11.88	1
Toll-like receptors	8	12.45	1
Transforming growth factor beta	17	25.92	1

## Data Availability

The datasets generated for this study can be found in the National Center for Biotechnology Information (NCBI) in Bioproject: PRJNA544778. The pacbio data: SAMN16560737 name: Coral_E4_day0 RNA-Seq of *M. foliosa*: polyps. Illumina HiSeq X: SAMN16560620; SAMN16560621; SAMN16560622. Transcriptome annotation data: SAMN16561030 name: Coral_E4_day0_Gene_Expression RNA-Seq of *M. foliosa*: polyps. All expression data and enrichment analysis results are showed in Appendix A.

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
