# Peer review of "Full-Length Transcriptome Sequencing of the Scleractinian Coral Montipora foliosa Reveals the Gene Expression Profile of Coral–Zooxanthellae Holobiont"

_biology, 2021, doi:10.3390/biology10121274_

Round 1

Reviewer 1 Report

The manuscript entitled “Full-length transcriptome sequencing of scleractinian coral Montipora foliosa reveals the gene expression profile of coral-zooxanthellae holobiont” by Liu et al. presents transcriptome data from long-reads and short-reads sequencing platforms to construct a reference transcriptome for the coral and zooxanthellae and identify putative genes and associated biological process related to the coral symbiosis.

Overall, the manuscript is technically sound and provides an important genetic resource for future investigations in the coral-zooxanthellae symbiosis. Therefore, I would recommend this MS for publication in the journal after addressing a few minor revisions

Abstract:

I would encourage the authors to modify the abstract as it contains mostly methods and leaves almost no insights about the results and the relevance of the results.

M&M

Lines 96 – 99: it Is important to highlight whether samples were taken multiple times from a single colony or correspond to different colonies. i.e., the samples repetition mentioned in this section corresponds to technical replicates or biological replicates.

Lines 182 – 188: It is not clear to me how did you “compared” the transcript data with the three genomes. Please provide more details and specify if you used the same software and parameter for all comparisons.

Line 193: “Illumina sequencing were compared to the REF”. Do you mean reads were mapped back to the REF?

Results:

Table 1: Check for column names

Figure 5: It called my attention to the bimodal distribution of the transcript’s length. Does this include both the coral and the zooxanthellae genes?

Discussion

Overall, this section feels more like a literature review about the molecular mechanisms involved in the symbiotic interaction between the coral and the zooxanthellae, and, at times, it is hard to relate the results from this study to the overall model described here.  Considering that the authors have the abundance calculation, it would be helpful to bring the abundance of the genes involved in the model described in Fig.8 into the main body of the manuscript and not just as supplementary material.

Lines 394 – 396: But as far as I understood, these genes were highly expressed in the coral, not the zooxanthellae. Which genes are involved in this process? Maybe they also take in another more relevant function.

Lines 432 – 441: Who proposed this model? Include references.

Author Response

Dear Sir:
Please check the response in the attachment file.

Yunqing Liu

Reviewer 2 Report

Review

Paper title: Full-length transcriptome sequencing of scleractinian coral Montipora foliosa reveals the gene expression profile of coral-zooxanthellae holobiont.

For the first time, the authors conducted a laboratory transcriptome sequencing study to obtain high-quality transcripts of the scleractinian coral Montipora foliosa and summarized information regarding their functions. The authors identified different isoforms of Montipora foliosa transcripts. The authors studied gene expression in the coral-zooxanthellae system. The authors compared their results with previous studies and discussed them in regard to physiological functions.

All these reasons explain the relevance of the paper by Yunqing Liu and co-authors submitted to "Biology".

General scores.

The data presented by the authors are original and significant. The study is correctly designed and the authors used appropriate methods. In general, the statistical analyses are performed with good technical standards. The authors conducted careful work that may attract the attention of a wide range of specialists focused on transcriptome analysis of aquatic organisms.

Recommendations.

L 26-29. The authors should avoid citations in this section. In exceptional cases, citations can be used but in the full form.

Section 2.3.2. The second part of this text is part of the manufacturer’s instructions (L 116-123). The authors should modify it to a standard description of Materials and Methods as they did in previous sections.

L 213. The authors should provide a more detailed table caption,

L 307. The authors should include Figure 5 in the paper.

L 432-441. The authors should provide citations for this section.

L 658-668. This section includes a mix of results and discussion. The authors should separate the text and present the results in the corresponding section.

Specific comments.

L 2. Change “scleractinian” to “the scleractinian”

L 13. Change “understanding” to “an understanding”

L 44. Change “extract the” to “extract”

L 45. Change “provide” to “provides”

L 58. Change “algae  is” to “algae  are”

L 70. Change “current published” to “currently published”

L 182. Change “public database” to “a public database”

L 200. Change “literature” to “the literature”

L 218. Change “remove redundant of transcript” to “removal of redundant transcripts”

L 222. Change “length frequency” to “length-frequency”

L 223. Change “in the Tables” to “in Tables”

L 227. This sentence “The databases used…” is a repetition of previously mentioned information. 

L 247. Change “was annotated” to “were annotated”

L 254, 266. Delete “results”

L 271, 272, 281. Change “endoplasmic reticulum” to “the endoplasmic reticulum”

L 293. Change “Pfam database” to “Swiss-Prot”

L 302. Change “Statistics results” to “Statistical results”

L 304, 306, 330, 354. Delete this sentence “The abscissa represents…”

L 353. Change “process, cellular component, and molecular function” to “processes, cellular components, and molecular functions”

L 364. Revise “abundant and depleted” (abundant or depleted? depleted?)

L 387. Change “located in” to “located”

L 389. Change “Research show” to “Research shows”

L 424. Change “is recognition” to “are recognition”

L 469. Change “originate” to “originates”

L 471. Change “sequentially” to “are sequentially”

L 473. Change “its biological functions” to “their its biological functions”

L 477. Change “components. Such as C1 to C9” to “components such as C1 to C9”

L 485. Change “The main activator of the classical pathway is antibody” to “The main activators of the classical pathway are antibodies”

L 500. Change “system plays” to “system play”

L 507. Change “is one of the activation pathways of the complement system and is” to “are one of the activation pathways of the complement system and are”

L 512. Delete “including”

L 513. Change “involved” to “are involved”

L 519. Change “widely present” to “is widely present”

L 523. Change “has been replicated” to “have been replicated”

L 526. Change “In mouse” to “In mice”

L 529. Change “Three functions of properdin in vertebrates that may be related to symbiosis include” to “In vertebrates, three functions of properdin that may be related to symbiosis include”

L 531. Change “can as” to “can act as”

L 533. Change “participate” to “participates”

L 536. Change “is considered” to “it is considered”

L 545. Change “belonging” to “belongs”

L 547. Change “referred” to “referred to”

L 560. Change “These gene are belonged” to “These genes belong”

L 571. Change “glutamate” to “we found glutamate”

L 589. Change “found a” to “found an”

L 602. Change “participate” to “participated”

L 613. Change “TLR involved” to “TLR is involved”

L 615. Change “TLR protein” to “TLR proteins”

L 627. “Mus musculus” should be italicized.

L 628. Change “have two isoforms” to “has two isoforms”

L 657. Change “zooxanthellae” to “zooxanthellae which”

L 679. Change “a valuable of” to “valuable”

Author Response

Dear Sir:
Please check the response in the attachment file.

Yunqing Liu

This manuscript is a resubmission of an earlier submission. The following is a list of the peer review reports and author responses from that submission.

Round 1

Reviewer 1 Report

My suggestions:

  1. In discussion, I would add a possible pathway figure on coral-zoxantellae symbiosis.
  2. In the part of complement and receptor, could you give a few examples in discussion? I would rather mention the significant complement related genes.
  3. In chapter "Metabolism-related genes" I would mention a few examples on significant genes. 
  4. I would highlight the importance and need for reference genome in coral species.
  5. Were there any similar studies performed on different coral species?

Reviewer 2 Report

The manuscript by Liu and colleagues investigates, by transcriptomics means, the coral-zooxanthellae symbiosis in Montipora foliosa. Aside from the use of PacBio sequencing to obtain full-length transcripts, which is interesting by itself and quite innovative as far as corals are concerned, this study unfortunately suffers from severe issues that in my opinion fervent further consideration for publication at this stage.

First, the manuscript is written in a style which would be best suited for a textbook, rather than for a scientific study. Several unnecessary explanations are present and the text should be significantly trimmed. As far as the methodologies are concerned, the method used for the taxonomic assignment of transcripts to corals or zooxanthellae was not convincing at all. The authors will need to improve this point, exploiting available genomic data for Montipora spp. and Symbiodinium spp. to define the origins of each transcript.

The discussion is nearly non-existent. The authors just report very broad functional categories, apparently with very little knowledge of the evolution of given gene families, mixing up data available for vertebrates and invertebrates. Overall, the discussion section does not add much to the text and in summary I feel like this section fails by far to provide interesting insights in the main findings of this study.

Abstract; “relatively reliably” does not make any sense. Please revise

Montipora foliosa should be in italics. Please check and correct throughout the manuscript.

Line 26: “uniform transcripts” has no meaning

Lines 69-71: the quality of the English language used here should be improved. It is quite unclear what the authors mean by “lack of complete reference genome data”, as some coral genomes that could be considered “complete” are available. Plus, “shortage” is not an appropriate term to indicate the short length of Illumina reads.

Line 76: “to find the isoform of each gene” should be replaced by “to study alternative splicing”

Line 99: establishing -> revise with a more appropriate word

Line 110: “after polymer read bases were performed” -> clarify

“original offline data” -> unclear

Section 2.4.3 is somewhat redundant and should be joined with the previous one

Line 163-173: the text is unclear, at times confusing; in general the quality of the English language significantly declined in the materials and methods section.

Lines 205-208: once again, the text is unclear. First, FPKM is not an appropriate metric for between-samples comparisons. TPMs should be used instead. Then, it is not clear which comparisons were made, considering that no explanation was provided so far concerning the experimental design.

Lines 210-226: most of this text is unnecessary. This is not a textbook and readers are expected to know how such annotations resources are organized and to be able to retrieve these data from the references provide.

Table 1: Explain what does “G” stands for. The readers are not supposed to guess what the authors mean.

Section 3.2 is unnecessary. As said above, this is not a textbook, nor a user manual

Section 3.3 contains once again redundant information. CD-HIT has been mentioned, along with its use, in the materials and methods section. Just summarize the results in a table.

Figure 1 and text: I don’t really think the approach used for the discrimination between coral and zooxanthellae transcript was appropriate. Both Montipora spp. and Symbiodinium spp. genomes are available, hence a rather simple approach based on the comparative evaluation of e-values obtained against the two would have been sufficient for a reliable taxonomic assignment. The materials and methods section was definitely not clear about the strategy used.

Section 3.4, 3.5 and subsections: each of these contain unnecessary comments about the contents and usefulness of different annotation databases. This would be appropriate for a textbook, not for a scientific paper.

Talking about “successful” annotation is not appropriate.

Line 367: the authors probably mean SSR, not CDS here. In any case, this section is disjoined from the rest of the text and it is not clear why the authors think reporting SSR data is relevant in this case.

The results presented in 3.6.1 would make sense, assuming that the genes identified as belonging to Symbiodinium are correct. However, based on the poor clarity of the methodology, it is difficult to evaluate whether this was the case or not.

The organization of the discussion section is not appropriate, as it just appears to be a list of genes belonging to different categories, with very little discussion about their potential role and, in particular, no inference made whatsoever concerning their possible involvement in symbiotic processes based on expression levels.

“Complement and its receptor” does not seem to apply to Cnidaria. Several studies have been published on the subject and the authors seem to ignore that only a very basic proto-complement system was likely present in the latest common ancestor of Cnidaria and all other metazoans, whereas the complement system in the form it is described here was only developed much later. Hence, we don’t really expect to find most of the components of the complement system in corals.

TLRs: same as above. The authors did not provide a reliable overview of the differences between this pathway in vertebrates and invertebrates. Talking about “TLR6” does not make much sense, as TLR numbering refers to vertebrate TLRs, which are definitely not homologous to those found in invertebrates.

4.2. “Cell-member related”?

This category is so broad and functionally diversified that this section is not useful at all. The same applies to 4.3 and 4.4

Reviewer 3 Report

In the current manuscript, the authors analyzed the gene expression profile of the scleractinian coral Montipora foliosa using PacBio PacBio Sequel II sequencing technology polished with the Illumina RNA‐seq platform. The identified transcripts were annotated in seven databases. The symbiosis-related genes were identified, and GO and KEGG pathways were enriched and studied. In general, the study provides important information on the gene expression profile of Montipora foliosa, and provides a good background for future research on symbiosis of this species and other related species. However, the manuscript may not be suitable for publication in its current state. Below are comments and suggestions for the authors. 

Major comments:

Materials and Methods:

How many biological and technical replicates were used for the current study beginning with culturing the coral samples until the completion of sequencing. This critical information is missing.

Please provide additional information on the methodology of Illumina sequencing or provide a reference so that the methods can be easily replicated. 

Results:

This is a general comment for the results section. Several paragraphs of the results contain information that does not belong to the results section. This information is important and provides necessary background information but it should be included in the introduction or discussion, wherever relevant. Please see my comments below on this point. 

General comment for the figures:

The figures should stand alone, therefore, I highly recommend defining all the abbreviations in the figure legends. Some figures need larger font size so that they can be easily read. Please see my comments below.

Other comments:

Line 27: (NR, NT, …). The abstract should stand alone, therefor I suggest defining the database abbreviations. The same comment for LncRNA (line 29).

Also, please clarify the species here (lines 80 and 83).

Line 106: …… were used ….

Lines 125-126: CCS has been already defined before (line 112). No need to define it here.

Lines 174-198: Abbreviations should be defined in their first occurrence. Example: CNCI should be defined in line 174 not 175, … etc. Please define LncRNA also.

Line 191: Please define SSR.

Line 216: …….. Molecular Signatures DataBase (MSigDB) software [35] to …….

Line 217: GO has bee already defined before.

Table 1: What is Title 2? Definition of (G)? I suggest deleting “Statistical” from the title.

Line 256: …. NR, NT, …… Also Line 263 (NR).

Figure 1: The font size of the database names in A and B are too small. In C and D, Nr should be NR.

Lines 278-279: The first two sentences beginning with “NR is called non‐redundant protein ….” are not results.

Lines 286-295: This paragraph contains important description od the GO database but most of it (The full name …….. metabolism) does not belong to the results section. It can be moved to the introduction. I suggest moving the description of other databases also from the results to a single paragraph in the introduction.

Figure 2: Legend: here GO and MF should be defined so that the figure can stand alone. The same for KEGG and MF in Figure 3. Also, abbreviations for Figure 4, 5, and 6 should be defined.

Figure 5: I suggest enlarging the figure or at least increase the font size= so that it is easier to read. To be consistent, I suggest using uppercase letters for the abbreviations such as cnci, … etc.

Lines 301-309 and 313- 320: Please see my comment on lines 286-295. Also, lines 329-336, 340-342. I suggest combining all the results of “Gene function annotation” under a single title and move the introductory database information to the introduction, and use relevant information in the discussion section.

For 3.5 Gene structure analysis and all the subheadings: Several paragraphs begin with information about the title being presented. Again, this is beneficial to the reader as it provides the necessary background information to understand the results. However, I highly recommend that this information be moved to the introduction or the the discussion wherever is more relevant. The results should mainly include what was discovered in the study. Also, 3.6 Gene expression analysis is a mixture of results and discussion (mostly discussion).

Line 418: VLKP definition?

Discussion: I suggest beginning the Discussion with a one-paragraph summary of the major findings. Also, I suggest removing all the subheadings under the Discussion.

Citations in the main text: I suggest using two square brackets for double citations (example, line 46 should be [3,4] not [3][4]) and a range for multiple citations (example, line 58 should be [6-8] not [6][7][8]).

Author contributions: Looks like this section is missing and only text from Biology template is there.

References:

The title of cited papers should be in the same letter case. Example, most references are in the “Sentence case” while for some others (#4, 11, … etc), Each Word Is Capitalized. For # 50 and 52, the title is “ALL CAPS”.

Reference # 18: The authors cited a preprint that have not been peer reviewed. Please replace it with the peer reviewed paper (Nature Methods 13, 1050-1054 (2016).

Round 2

Reviewer 1 Report

Authors fulfilled my suggestions. Thank you.

Reviewer 2 Report

While I appreciate the efforts made by the authors to improve their manuscript in such a short amount of time, I am sorry to say that the modifications made are in my opinion still largely unsatisfactory.

Lines 221-281: this is an example of what I meant by saying that this MS looks like a textbook. Any reader is supposed to know what a protein domain is, what a CDS is, what KEGG, GO and COG are. It is not a question of redundancy in the text, it is a question of providing unnecessary information to the readers.

2.4.4. This is still largely unsatisfactory. Taxonomical classification cannot be simply made based on keywords: it needs to be based on objective criteria and (in the case of BLAST) e-value thresholds. As I have previously said, plenty of genomic information are available, bot for corals and for zooxanthellae. Not just in the NR database, but in the shotgun genome assembly database, which includes much more data than NR. These should have been used to provide a reliable transcript classification.

I appreciate the efforts made by the authors n trying to improve the discussion. However, since there is currently no proof that the taxonomical assignment of transcripts was done properly, unfortunately it is hard to evaluate whether the data contained in this section are appropriate or not.

About the complement system: the updates made are not satisfactory. The authors should have, to the very least, mentioned the existence of the classical, alternative and lectin pathways, explaining which components were likely to be present in the proto-complement system found in non-deuterostome invertebrates. Once again, it does not make much sense to report very remote sequence similarities with complement receptors in metazoans where no complement receptors are expected to be found. Note that some BLAST annotations may be false positives.

With this respect, the discussion about lectins is also extremely simplistic. The authors fail to show why TSR proteins should be considered as important in this symbiosis.

The authors have ignored my suggestions about TLRs.

Reviewer 3 Report

I would like to thank the authors for diligently revising their manuscript and addressing the comments. This version has been significantly improved.